# Genome-Wide Identification and Characterization of the *HMGR* Gene Family in *Taraxacum kok-saghyz* Provide Insights into Its Regulation in Response to Ethylene and Methyl Jsamonate Treatments

**DOI:** 10.3390/plants13182646

**Published:** 2024-09-21

**Authors:** Pingping Du, Huan He, Jiayin Wang, Lili Wang, Zhuang Meng, Xiang Jin, Liyu Zhang, Fei Wang, Hongbin Li, Quanliang Xie

**Affiliations:** Key Laboratory of Xinjiang Phytomedicine Resource and Utilization of Ministry of Education, Xinjiang Production and Construction Corps Key Laboratory of Oasis Town and Mountain-Basin System Ecology, College of Life Sciences, Shihezi University, Shihezi 832003, China; dopingping@126.com (P.D.); he_huan026@163.com (H.H.); wjyinee@163.com (J.W.); lfyroquz@163.com (L.W.); zhuangmeng610@163.com (Z.M.); jinx@hainnu.edu.cn (X.J.); zhangliyu981125@163.com (L.Z.)

**Keywords:** *HMGR*, *Taraxacum kok-saghyz*, natural rubber biosynthesis, ethylene, methyl jasmonate

## Abstract

*HMGR* (3-hydroxy-3-methylglutaryl-CoA reductase) plays a crucial role as the first rate-limiting enzyme in the mevalonate (MVA) pathway, which is the upstream pathway of natural rubber biosynthesis. In this study, we carried out whole-genome identification of *Taraxacum kok-saghyz* (TKS), a novel rubber-producing alternative plant, and obtained six members of the *TkHMGR* genes. Bioinformatic analyses were performed including gene structure, protein properties, chromosomal localization, evolutionary relationships, and cis-acting element analyses. The results showed that *HMGR* genes were highly conserved during evolution with a complete HMG-CoA reductase conserved domain and were closely related to *Asteraceae* plants during the evolutionary process. The α-helix is the most prominent feature of the secondary structure of the TkHMGR proteins. Collinearity analyses demonstrated that a whole-genome duplication (WGD) event and tandem duplication event play a key role in the expansion of this family and *TkHMGR1* and *TkHMGR6* have more homologous gene between other species. Cis-acting element analysis revealed that the *TkHMGR* gene family had a higher number of MYB-related, light-responsive, hormone-responsive elements. In addition, we investigated the expression patterns of family members induced by ethylene (ETH) and methyl jasmonate (MeJA), and their expression levels at different stages of *T. kok-saghyz* root development. Finally, subcellular localization results showed that six TkHMGR members were all located in the endoplasmic reticulum. In conclusion, the results of our study lay a certain theoretical basis for the subsequent improvement of rubber yield, molecular breeding of rubber-producing plants, and genetic improvement of *T. kok-saghyz*.

## 1. Introduction

Plant secondary metabolites are crucial for plant growth, development, defense, and regulation, as well as a range of important natural products [1]. Natural rubber (NR), cis-1,4-polyisoprene, is a secondary metabolite of rubber-producing plants and an irreplaceable high-molecular-weight biopolymer. It has physical properties such as high elasticity, wear resistance, and impact resistance and has become a vital raw material in transportation, medicine, and other industries [2,3]. Currently, with the rapid development of economic demand, the demand for NR is increasing, and all countries in the world are facing the problem of NR shortage [4,5]. More than 2500 plant species produce NR, but the rubber tree (*Hevea brasiliensis Muell. Arg.*) is the only commercially viable rubber producing plant that yields high-quality NR [2,5]. However, due to the vulnerability of rubber producing systems, which are prone to tapping panel dryness (TPD) (15–50%), about 15–20% of NR production is affected every year [2,3,4,5,6]. Genetic improvement in *H. brasiliensis* is both slow and time-consuming [7]; therefore, seeking new rubber producing substitute plants provides a way to alleviate the shortage of NR.

*Taraxacum kok-saghyz* L. *Rodin* (TKS), was discovered and brought to attention in the 1930 s [8]. It is a perennial herb belonging to the *Taraxacum* genus in the family of *Asteraceae* [9], whose roots are the main rubber-producing part and can produce high-molecular-weight rubbers ranging from 2.89% to 27.89% [8]. Compared with rubber trees, TKS has the advantages of easy genetic transformation [10], such as shorter growth cycles, and the production of other useful secondary metabolites [3]. So, it has been widely studied as a model plant for the natural rubber biosynthesis (NRB) mechanism in recent years [11]. NR is formed by the continuous production of isoprene pyrophosphate (IPP) monomers in the cytoplasmic mevalonate (MVA) pathway and plastid 2-C-methyl-D-erythritol-4-phosphate (MEP) pathway [12]. The IPP monomer is the basic unit of NR. Under the joint action of the complex formed by cis-prenyltransferase (CPT), rubber elongation factor (REF), small rubber particle protein (SRPP), and HRT1-REF bridging protein (HRBP), IPP monomers are polymerized on the rubber particle membrane to form IPP long-chain molecules. The IPP long-chain molecules are then dissociated from the rubber transferase complex to form rubber molecules and are stored in the rubber particles [3,13]. It has been shown that the MVA pathway is the main contributor to IPP and the most important synthetic pathway of NR [14].

HMGR (3-hydroxy-3-methylglutaryl coenzyme A reductase) is the first rate-limiting enzyme in the MVA pathway, catalyzing the NADPH-dependent synthetic reaction from 3-hydroxy-3-methylglutaryl coenzyme A to mevalonate and controlling the reaction rate of the MVA pathway [15,16]. HMGR was purified in 1986 after its discovery in 1958 [17]. It has been identified as a membrane-bound hydrophobic protein, and the sequence of HMGR in plants is characterized by four types structural domains: a variable cytoplasmic N-terminal structural domain, two transmembrane structural domains spanning the endoplasmic reticulum, a variable regionally linked transmembrane structural domain, and a highly conserved cytoplasmic C-terminal catalytic structural domain [18,19]. Within the catalytic domain, there are two HMG-CoA-binding motifs (EMPVGYVQIP and TTEGCLVA) and two NADP(H)-binding motifs (DAMGMNM and GTVGGGT), which are important for the function of the HMGR protein [20]. Not only that, HMGR is a highly conserved enzyme with homologous sequences in eukaryotes, prokaryotes, and archaea [21]. There are two classes of HMG-CoA reductase-conserved domains (CDD accession: cl00205). Class I (CDD accession: cd00643) is mainly found in eukaryotes and is highly similar to human HMGR; it contains an N-terminal membrane region and utilizes NADPH as an electron donor. Class II (CDD accession: cd00644) is mainly found in prokaryotes without a membrane region and utilizes NADH as an electron donor [21,22,23]. It is the highly conserved sequence of the HMGR protein that makes it also play a significant role in adapting to changing environments.

The gene encoded by *HMGR* was first cloned in *Arabidopsis thaliana* in 1989 [17,24]; so far, it has been isolated and cloned from more than 80 species of different plants such as rice [25], cotton [26], wheat [27], the rubber tree [28], and so on. However, the regulatory function of the *HMGR* gene in TKS is rarely reported. The *HMGR* of *H. brasiliensis* was transferred into tobacco for heterologous expression, and the HMGR enzyme activity of the obtained transgenic plants was increased four~eight times, and the total sterol content was increased six times [29]. Overexpression of the *HMGR* gene in the rubber tree resulted in a positive correlation between the latex yield of the transgenic rubber trees and the HMGR enzyme activity [4]. Heterologous overexpression of three key enzymes (AACT, ACT and HMGR) of the *A.thaliana* MVA pathway in *Taraxacum brevicorniculatum* results in the transgenic lines with higher levels of sterols, pentacyclic triterpenes, and cis-1,4-isoprene [30]. Overexpression of the *PgHMGR* can significantly increase ginsenoside content in ginseng [31]; overexpression of *PtHMGR* improves stress tolerance in poplar [32]. Potato *HMGR1* is mainly involved in sterol synthesis, and *HMGR2/3* mainly affects sesquiterpene accumulation [33]. In *Vitis vinifera*, *VvHMGR3* plays a significant role in the process of fruit coloring and aroma formation [34]. In apples, various hormone-responsive elements were found in the promoters of *MdHMGR1*, *MdHMGR2*, and *MdHMGR4*, under different hormone treatments such as ethylene (ETH), methyl jasmonate (MeJA), and salicylic acid (SA), their expression patterns showed different trends, with relatively high expression in roots and stems [35,36]. These results indicate that *HMGR* promotes the synthesis of plant secondary metabolites, plant growth and development, and coping with abiotic stresses and environmental changes. However, the specific function of the *HMGR* gene of NRB in *T. kok-saghyz* remains unclear; the *HMGR* gene family in TKS has not been studied in depth. The relationship between *HMGR* and the cumulative change in rubber yield, the effect of *HMGR* on the secondary metabolites of TKS, and the possible interaction with other members of the rubber synthesis pathway to regulate rubber synthesis need to be further explored.

It was critical to identify candidate genes in the NRB pathway and use strategies such as transgenesis or synthetic biology as a means to increase rubber yield. In order to preliminarily investigate the effect of *TkHMGRs* on NR synthesis, the whole genome of TKS was used to identify the members of the *TkHMGR* gene family, and the members of the *TkHMGR* family were comprehensively analyzed in this study. We carried out a series of biological methods to explore gene structure, protein structure, collinearity, evolutionary relationships, and cis-acting elements; explored the expression pattern through qRT-PCR and RNA-seq; and investigated their functional location through subcellular localization. Our findings may provide a theoretical basis for subsequent functional verification experiments and molecular breeding. 

## 2. Results

### 2.1. Identification of TkHMGR Family Members in T. kok-saghyz

A total of six *TkHMGRs* were identified in the TKS genome, and the members were named *TkHMGR1*~*TkHMGR6* according to the order of localization in the chromosome. The predicted relevant physical and chemical properties are shown in Table 1 and Appendix A. The longest protein consisted of 614 amino acids (TkHMGR1), and the shortest protein consisted of 406 amino acids (TkHMGR3); molecular weight sizes ranged from 42.694 (TkHMGR3) to 66.68 kDa (TkHMGR1). The isoelectric points of the three members (TkHMGR2, TkHMGR4, and TkHMGR5) were significantly < 7, suggesting that they were acidic proteins. The aliphatic index of all the members fluctuated around 92, which is relatively consistent, and it is assumed that the thermal stability of the proteins is similar. The TkHMGR2, TkHMGR3, and TkHMGR4 proteins were more stable with an instability index of < 40, while the other three proteins were more unstable. The grand average of hydropathicity (GRAVY) of the proteins was > 0, except for TkHMGR5. Therefore, it is speculated that most of the TkHMGR proteins are hydrophobic. Subcellular localization prediction showed that all six proteins were localized in the endoplasmic reticulum. Prediction of the transmembrane regions of the proteins showed that TkHMGR1 has three transmembrane helices, and both TkHMGR2 and TkHMGR6 have two transmembrane helices (Appendix A); all of them span both inside and outside of the membrane.

### 2.2. Phylogenetic Relationship, Conserved Structure, Motif Composition, and Gene Structures of TkHMGRs

Based on the phylogenetic analysis, TkHMGR3 and TkHMGR5 are more closely related evolutionarily, as well as TkHMGR1 and TkHMGR6 (Figure 1A). Conserved structural domain analyses indicate that the TkHMGR proteins all have HMG-CoA reductases I (Figure 1B). Among the 10 predicted motifs, motif 1 contained one NADP(H) binding site, motif 3 contained two HMG-CoA binding sites, motif 9 contained one NADP(H) binding site, and all of them were highly conserved (Figure 1C,E). TkHMGR1, TkHMGR2, and TkHMGR6 all contain 10 motifs, while TkHMGR3~TkHMGR5 contain 9 motifs with the absence of motif 7. Gene structure analysis showed that *TkHMGR3*, *TkHMGR4*, and *TkHMGR5* had one intron; *TkHMGR1*, *TkHMGR2*, and *TkHMGR6* had three introns. *TkHMGR3*, *TkHMGR4*, and *TkHMGR5* contained two coding regions; *TkHMGR1*, *TkHMGR2*, and *TkHMGR6* contained four coding regions (Figure 1D). Differences in intron diversity can provide additional insights into the evolution of the *HMGR* gene family. *TkHMGR1*, *TkHMGR2*, and *TkHMGR6* possess longer coding sequences and have a higher number of exons with coding regions compared to other members, and these three genes may contribute to the functional diversity and specificity of *TkHMGR* genes. All the results suggest that the gene structures, exon-intron counts and gene lengths of the same class of proteins are basically similar, and the functions they perform may also be similar.

### 2.3. Multiple Sequence Alignments, Secondary Structure Analysis, and Multispecies Phylogenetic Analysis of HMGR Proteins in T. kok-saghyz

Multiple sequence alignment analysis of TkHMGR proteins revealed that each member contained two HMG-CoA binding sites and two NADP(H) binding sites (Figure 2). Two NADP(H) binding sites and one HMG-CoA binding site were not mutated or lost during evolution; there was one HMG-CoA binding site in which the conserved amino acid V (tryptophan) was mutated to I (isoleucine) (Figure 2). All these indicate that these binding sites of TkHMGRs are highly conserved during the evolutionary process. TkHMGR1 has three transmembrane helices, TkHMGR2 and TkHMGR6 have two transmembrane helices each, and the missing transmembrane helices are all TMH 1 (Figure 2), which is also confirmed by the prediction in Table 1. The secondary structure predicted via TkHMGR1 was shown above the multiple sequence alignments, with 15 α-helixs, 11 β-strands, four strict β-turns, and three 3_10_-helixs. 3_10_-helixs are specific helixs of the TkHMGR proteins. Corroborating with the results of predicting the secondary structure of proteins through the SOPMA website (Table 2), helix structures accounted for the highest percentage of this gene family, followed by random curl structures (where there is no tag in the multiple sequence alignment above), then strand structures, and the least was turn structures. In addition, the homology modeling method was used to predict the three-dimensional model of six TkHMGR proteins (Appendix A), and the structures among the members were similar, all of which indicated that the evolution of HMGRs in TKS was conservative.

To deeply explore the evolutionary relationship of *TkHMGRs* with other species, a phylogenetic tree was constructed by combining *Taraxacum kok-saghyz* and nine other species including *Hevea brasiliensis*, *Taraxacum mongolicum*, *Lactuca sativa*, *Nicotiana tabacum*, *Helianthus annuus*, *Arabidopsis thaliana*, *Oryza sativa*, *Eucommia ulmoides*, and *Zea mays* into three clusters (Figure 3). *L. sativa* possessed eight HMGR family members, whereas only two AtHMGRs were identified in *A. thaliana*. Most of the TkHMGR proteins clustered together with *H. annuus*, *L. sativa*, and *T. mongolicum* as Clade III, which is related to their belonging to the same family of *Asteraceae*. Clade I mainly includes HMGR members of maize and rice, both of which have a small number of members (3 or 4), and they are clustered together because of their close affinity to the same family of *Poaceae.* Clade II includes most of the non-*Asteraceae* plants, and TkHMGR proteins are not clustered with *H. brasiliensis* and *E. ulmoides*, which are the same plants that produce natural rubber. 

### 2.4. Chromosomal Localization and Collinearity Analysis in T. kok-saghyz

TKS is a diploid plant (2n = 16, n = 8); chromosome 1 is the longest, and chromosome 8 is the shortest (Figure 4). *TkHMGRs* are mainly distributed on chromosomes 2, 5, and 6, with the most members on chromosome 5, including *TkHMGR2*~*TkHMGR5*; *TkHMGR1* and *TkHMGR6* are located on chromosomes 2 and 6, respectively (Figure 4). Gene duplication events are crucial to genome and gene system evolution. Collinearity analysis of the *TkHMGR* gene family members shows that *TkHMGR1* and *TkHMGR6* have collinear relationships and may participate in whole genome duplication events, while *TkHMGR3*~*TkHMGR5* are tandem duplication genes (Table 3). In addition, an interesting finding is that *TkHMGR2* is also involved in genome-wide duplication events, and the collinear gene is *TkA03G47197.1*, but it does not belong to this gene family. 

The Ka/Ks values of two duplicate gene pairs were calculated using the non-synonymous replacement rate (Ka), synonymous replacement rate (Ks), and Ka/Ks ratio (Appendix A). It was found that their Ka/Ks values were far less than 1, indicating that they had all undergone strong purification selection. These findings provide a potential basis for revealing the evolutionary relationships in the *TkHMGR* gene family.

According to the species evolution relationship, the collinearity relationship between *T. kok-saghyz* and *A. thaliana*, *H. annuus*, *H. brasiliensis*, *L. sativa*, and *T. mongolicum* was established (Figure 5). There were three collinear relationships between *TkHMGRs* and *AtHMGRs* and *HaHMGRs*, four collinear relationships between *TkHMGRs* and *HbHMGRs*, six collinear relationships between *TkHMGRs* and *LsHMGRs*, and five collinear relationships between *TkHMGRs* and *TmHMGRs*. There were more homologous genes with *TkHMGR1* and *TkHMGR6*, and it was speculated that they play an important role in promoting the species diversity of the *HMGR* gene family. Genes homologous to *TkHMGR2* and *TkHMGR3* are only found in more closely related species. Interestingly, a homologous gene of the *HMGR* in *T. mongolicum*, *TmA02G104750.1* (sequence see Appendix A), was identified via collinearity analysis; it is homologous to *TkHMGR1* and *TkHMGR6* but is not a member of the *TmHMGRs*, and further studies have showed that it has lost important functional regions during evolution, an NADPH-binding motif (GTVGGGT).

### 2.5. Cis-Acting Element Analysis of HMGR Gene Family in T. kok-saghyz

To understand the potential functions of the *HMGR* gene family in TKS, six members were analyzed for cis-acting elements, and the results showed that a relatively abundant number of related elements were predicted, such as an MYB-related response element, light response-related element, hormone response-related element, stress-related element, plant growth and development-related element (Figure 6). In this gene family, the more typeelements were the light response-related elements (17 kinds) and hormone response elements (11 kinds) (Figure 6A). The most numerous were 90 phytohormone-responsive elements, followed by 74 MYB-related elements (Figure 6B), which reflected the importance of plant hormone elements to *TkHMGRs* to some extent. On the other hand, the white values inside the small box significantly indicated that the gene had a higher number of this element (Figure 6A). We found that *TkHMGR1* had seven MYC elements; *TkHMGR3* had seven MYB elements and seven MYC elements; *TkHMGR4* had seven MYB elements, six G-Box elements, and six ABRE elements; and *TkHMGR5* had five Box4 elements. *TkHMGR6* had the most MYC-element, with 10. These results reveal that the drought response of *TkHMGRs* may be mainly regulated by MYB and MYC transcription factors.

The predicted phytohormone elements in six members include abscisic acid-responsive element (ABRE, ABRE3a, and ABRE4), MeJA-responsive element (CGTCA-motif and TGACG-motif), gibberellin-responsive element (GARE-motif and P-box), salicylic acid-responsive element (as-1 and TCA-element), ethylene-responsive element (ERE), growth hormone-responsive element (TGA-element), and so on. The light-responsive elements were diverse but not widely distributed and additionally contained many elements responding to low temperature (LTE) and anaerobic induction (ARE) stresses and meristem expression (CAT-box) and endosperm expression (GCN4 motif) plant growth and development-related elements. All these results suggest that *TkHMGRs* may be involved in responding to phytohormone regulatory mechanisms.

### 2.6. The Expression Pattern Analysis of HMGR in T. kok-saghyz and Physiological Indexes

The *TkHMGR* gene family members possesses a high number of responsive hormone elements, including ethylene (ETH)- and methyl jasmonate (MeJA)-responsive elements. We explored the roots’ and leaves’ expression pattern of *TkHMGR* gene family members at 0 h, 3 h, 6 h, 12 h, and 24 h, by treating the roots of 6-month-old TKS using ETH and MeJA. 

Under ETH treatment, the expression of *TkHMGR1*, *TkHMGR2*, and *TkHMGR6* in the roots of TKS changed in a large range (Figure 7A). *TkHMGR1* was the fastest in response to ETH, reaching the highest expression at 3 h; then, the expression gradually declined with the prolongation of time. The expression of *TkHMGR2* fluctuated but did not exceed that of the control. *TkHMGR6* increased with the extension of the treatment time, reaching the peak at 12 h, and was then slightly down-regulated at 24 h. The expression of *TkHMGR3*~*TkHMGR5* showed a similar expression pattern (Figure 7A), with a sharp decline in expression at 3 h, after which they remained at a low expression level. This indicated that they were down-regulated under ETH treatment and did not respond positively to ETH. In the leaf expression profile under ETH treatment (Figure 7B), the expression levels of *TkHMGR2*~*TkHMGR6* responded rapidly and peaked within 3 h or 6 h but were gradually down-regulated with increasing treatment time.

Under MeJA treatment, both *TkHMGR1* and *TkHMGR6* showed an up-regulation trend, reaching a peak at 12 h, with a down-regulation at 24 h but still exceeding the control (Figure 7D). *TkHMGR2*~*TkHMGR5* showed a similar trend, with a gradual decrease in expression with the increase in the treatment time, and both reached a minimum at 12 h, with a slight up-regulation trend at 24 h but not exceeding the control. In the expression profiles of leaves, the expression patterns of *TkHMGR3*~*TkHMGR5* were similar, and the expression levels peaked at 12 h; the expression level of *TkHMGR1* reaches its peak at 3 h and gradually decreases thereafter; the expression level of *TkHMGR2* decreases continuously with increasing processing time; only the expression level of *TkHMGR6* increased by 5–15 times with the continuous increase in treatment time. (Figure 7E). 

Further, to understand the effects of these two exogenous hormones on TKS, we determined the changes in proline, soluble sugar, and chlorophyll in the same treatment. Under both treatments, proline content and soluble sugar content of TKS roots showed an increasing trend and peaked at 24 h; the chlorophyll content of TKS leaves tended to decrease.

We also explored the expression patterns of the *TkHMGR* gene family members during different developmental periods in TKS roots (Figure 7G). The expression level of *TkHMGR2* is a hundred times higher than that of other genes. *TkHMGR1* expression remained relatively stable after a 1.5-fold increase in 3-month-old roots. The expression of *TkHMGR4* and *TkHMGR5* was low in 1-month-old plants and then gradually increased with plant growth. *TkHMGR3* expression in plants at 3 and 6 months of age had a decrease of about 4-fold, followed by an increase of 3~5-fold. Only the expression of *TkHMGR6* decreased continuously with root development. 

### 2.7. Subcellular Localization Analysis of Six TkHMGR Proteins 

To further explore the functions of TkHMGR proteins, subcellular localization was performed to verify the specific locations where they exert their functions. The gene *TkHMGR-pCAMBIA1300-eGFP* and endoplasmic reticulum marker were co-infected and expressed instantly in *Nicotiana benthamiana*. The endoplasmic reticulum marker was created by combining the signal peptide of *AtWAK2* at the N-terminus of the mCherry and the HDEL (his-asp-glu-leu) endoplasmic reticulum retention signal at the C-terminus. In order to further demonstrate co-localization, quantitative analysis was conducted by selecting the ROIs (regions of interest) in the merge field and analyzing the intensity and overlap of fluorescence. The results showed six members were all located in the endoplasmic reticulum, as revealed via the green fluorescence of the TkHMGR-pCAMBIA1300-eGFP fusion protein and the red fluorescence of the endoplasmic reticulum marker in the merge field (Figure 8), consistent with the results predicted based on previous sequence analysis (Table 1).

## 3. Discussion

HMGR (3-hydroxy-3-methyl glutaryl coenzyme A reductase) catalyzes the rate-limiting reaction of cholesterol biosynthesis in animals and is one of the most regulated enzymes. Lovastatin is a specific inhibitor of HMGR, which can bind to the active site of HMGR and competitively inhibit the transformation of HMG-CoA into mevalonate [38,39]. As the research on *HMGR* in plants becomes more and more extensive and in-depth, it has been identified and cloned in a variety of medicinal plants. The MVA pathway is an upstream part of NR synthesis, and regulation of HMGR may have an impact on NR production. In this study, we identified a total of six *TkHMGR* genes from the published whole genome of TKS by Lin et al. [37], and the physicochemical properties, structural functions, and evolutionary relationships of these members were analyzed (Table 1). The six genes identified through blastp analysis and determination of conserved domains method contradict the eight genes identified by Lin et al. in 2022 [37] (Appendix A). Through further confirmation, all the genes we identified are included in Lin’s identification (Appendix A). The other two genes (*TkA03G471970* and *TkA04G128060*) were not identified in this study because they were shown as pseudogenes in genome annotation and their protein sequences were not found in the proteome (Appendix A). And during the analysis of collinearity, we found one of the genes (*TkA03G47197.1*) (Figure 3) has a collinear relationship with *TkHMGR2*, which experience genome-wide replication events together. Pseudogenes have a similar structure to functional genes but have lost their coding ability to function as normal genes in evolution [40], and there are also reports that the pseudogenes have been found in the cotton *HMGR* gene family [41].

In this study, we found that all members of the *HMGR* gene family in *T. kok-saghyz* were located on the endoplasmic reticulum, which was the same as that predicted for the *HMGR* family in *Gossypium* [41], *Lithospermum erythrorhizon* [42], and *Vitis* [34], and that *TkHMGR1*, *TkHMGR2*, and *TkHMGR6* had trans-endoplasmic reticulum membrane helices (Table 1). In eukaryotes, the endoplasmic reticulum is a sorting site for proteins, and most of the proteins integrated into the endoplasmic reticulum membrane are α-helices with transmembrane helical structural domains [43], and the secondary structure of HMGR proteins is dominated by α-helices [44], which was also verified in this study (Figure 2, Table 2). In mammals, the HMGR protein is a transmembrane integrative glycoprotein residing in the endoplasmic reticulum and the only endoplasmic reticulum-located enzyme in the MVA pathway. It can control the rate of the MVA reaction, and when exogenous cholesterol is not sufficiently supplied, it increases MVA pathway activity to produce the necessary sterols and terpenes [45]. Thus, the transmembrane helix is an important feature of the HMGR. Multiple sequence alignments of TkHMGR proteins suggest to some extent that the longer the protein sequence, the greater the number of transmembrane helices (Figure 2). 

Gene family duplication events play a crucial role in rapid gene family amplification and evolution, providing opportunities for gene family expansion and diversity, providing new genetic material and opportunities to acquire new functions [46]. In this study, three segmental duplicated genes and three tandem duplicated genes were detected in the genome of TKS. WGD and tandem events play a major driving role in the expansion of the whole gene family. Phylogenetic analyses with the other nine species revealed that *L. sativa* had the highest number of members, whereas the minimum number of members in *A. thaliana* was only two, and TkHMGRs mostly clustered together with *H. annuus*, *L. sativa*, and *T. mongolicum*, which may be related to their belonging to the same family, *Asteraceae*. In the evolutionary process, the gene structure and protein structure of plant HMGRs are highly conserved, which is probably derived from an ancestor gene and eventually developed into two groups in monocot and dicot plants, respectively [20]. The evolution of HMGR from lower plants to higher plants is conserved [42]. We also found that the HMG-CoA reductases I conserved domain of six TkHMGRs remained basically intact during evolution. Further collinearity analyses revealed that more genes were homologous to *TkHMGR1* and *TkHMGR6*, and the number of *HMGR* collinearity was higher in species with closer affinities. 

Phytohormones are several trace compounds produced in plants that play a key role in plant growth, development, and response to biotic and abiotic stresses [47]. In this study, a large number of light-responsive and phytohormone-responsive elements were revealed by predicting cis-acting elements in the upstream region of the promoters of *TkHMGR* genes. In the regulation of ginsenoside biosynthesis by HMGR, it was found that light may down-regulate the expression of *HMGR* through the regulation of photoreceptor phytochrome B (phyB) and the transcription factor HYPOCOTYL5 (HY5) [31]. *VvHMGR3* was induced to be significantly up-regulated by ETH, IAA, SA, and MeJA [34]. On the other hand, ETH has been widely used to accelerate plant growth and development, including being used as a stimulant to increase NR in rubber tree [48]. In this study, the expression patterns of *TkHMGR* gene family members were investigated by treating the tissues of TKS with ETH and MeJA, and it was found that the expression levels of *TkHMGR1* and *TkHMGR6* were significantly up-regulated in roots of TKS. Under MeJA treatment, the leaf expression levels of *TkHMGR3*~*TkHMGR6* were significantly up-regulated, especially *TkHMGR6*. An interesting finding is that the expression pattern of *TkHMGR6* in roots and leaves is significantly up-regulated under MeJA treatment. In a previous study, He et al. assessed changes in the content of MDA (malondialdehyde) and SOD (superoxide dismutase), POD (peroxidase), and CAT (catalase) enzyme activity under ETH and MeJA treatments [49]; our study measured the changes in proline content and soluble sugar content. Accumulation of proline and soluble sugars are physiological response triggered by biotic or abiotic stress in plants [50,51,52]; their levels to some extent reflect the stress resistance of plants [50,52], and except for NR, inulin is another important soluble sugar product of TKS roots [37]. Under exogenous hormone stress, the proline content of TKS increased 2–3 times at 24 h, and the soluble sugars content increased 2 times at 24 h (Figure 7C,F). We also measured leaf chlorophyll content, which reflects physiological changes in the plant [53]. Chlorophyll content did not change much during the six hours of treatment, after which the content decreased significantly (Figure 7C,F). These findings may provide some insight into TKS breeding.

We also investigated the endogenous expression levels of *TkHMGRs* at different root development stages. The NR content of TKS aged six months can reach around 5.66% [54]; at the six-month mark of root development stages, the expression levels of *TkHMGR1*, *TkHMGR2*, and *TkHMGR6* were significantly higher than other genes. We also found that the endogenous expression levels of *TkHMGRs* are generally high in roots, especially *TkHMGR2*; this is consistent with what Lin et al. discovered [37].

HMGR function has been studied in *A. thaliana*, where post-translational regulation of HMGR may be related to protein phosphatase 2A [55], and the phosphorylation state of AtHMGR1S at the Ser577 site is important for regulating HMGR activity in *A. thaliana* [56]. Reversible phosphorylation of conserved sites in the catalytic structural domain is prevalent in plant HMGRs [57], and these phosphorylation modifications are important for regulating HMGR activity. However, the function of HMGR in TKS has not been thoroughly investigated, our studies could provide ideas for subsequent research to verify the function of HMGR. 

## 4. Materials and Methods

### 4.1. Genome-Wide Identification of HMGR Genes in T. kok-saghyz

To identify *TkHMGR* gene family members from the published genome of *T. kok-saghyz* [37], firstly, download the AtHMGRs and the TkHMGRs that are identified from NCBI as seed sequences. Then, use the blastp function of local blast to query candidate proteins among *T. kok-saghyz* protein sequences, removing members with similarity below 70%. Secondly, the raw Hidden Markov Model PF00368 for the HMGR protein conserved domain was downloaded from the Pfam database (http://pfam-legacy.xfam.org/, accessed on 9 October 2023) [58], and the hmmsearch function of HMMER v3.1b2 (http://hmmer.org/, accessed on 9 October 2023) [59] software was executed among candidate sequences. Then, the Batch CD-Search (https://www.ncbi.nlm.nih.gov/Structure/bwrpsb/bwrpsb.cgi, accessed on 9 October 2023) was utilized to remove redundant sequences that do not have HMG-CoA reductase class I domain (CDD accession: cd00643) with default parameters. Finally, six *TkHMGRs* were obtained [23]. The physical and chemical properties of the TkHMGR proteins such as protein length, molecular weight (MW), theoretical isoelectric point (pI), instability index, aliphatic index, and the grand average of hydropathicity (GRAVY) index were predicted via the Expasy website (https://web.expasy.org/protparam/, accessed on 14 October 2023) [60]. Subcellular locations of the TkHMGR proteins were predicted with Cell-PLoc 2.0 (http://www.csbio.sjtu.edu.cn/bioinf/Cell-PLoc-2/, accessed on 14 October 2023) [61]. In addition, the number of transmembrane helices of TkHMGR proteins were analyzed using the TMHMM website (https://services.healthtech.dtu.dk/services/TMHMM-2.0/, accessed on 14 October 2023). 

### 4.2. Phylogenetic Analysis, Gene Structure, Motif, and Conserved Domain Analysis 

The phylogenetic tree was constructed by using MEGA 11.0 with the maximum likelihood (ML) method with 1000 replications [62], to explore the evolutionary relationship of TkHMGRs. The conserved domains and conserved motifs of the six TkHMGR proteins were analyzed via the batch CD-Search and MEME program (Version 5.5.5) (http://meme-suite.org/tools/meme, accessed on 20 October 2023) [23,63], and the maximum number of motifs was 10. CDS sequences of *TkHMGRs* were obtained in the TKS genome database and then the Gene Structure Display Server (http://gsds.gao-lab.org/, accessed on 20 October 2023) [64] was used to predict the gene structure. Ultimately, the Evolview website (https://www.evolgenius.info/evolview/#/login, accessed on 20 October 2023) [65] was used to display the phylogenetic tree, gene structure, motif composition, and conserved domain. 

### 4.3. Multiple Sequence Alignments, Secondary Structure Prediction, and Protein 3D Structure Prediction

The muscle function of MEGA 11.0 was used to align the TkHMGR proteins, then ESPript 3.0 (https://espript.ibcp.fr/ESPript/ESPript/index.php#, accessed on 25 October 2023) was imported for display [66]. The secondary structure of the remaining proteins was predicted from the SOPMA website (https://npsa-pbil.ibcp.fr/cgi-bin/npsa_automat.pl?page=npsa_sopma.html, accessed on 25 October 2023). The 3D structure of proteins was predicted by using the swiss model (https://swissmodel.expasy.org/, accessed on 4 November 2023) [67].

### 4.4. Collinearity Analysis and Cis-Acting Element Analysis

The GC content, gene density, and chromosomal localization of the *TkHMGR* gene family were extracted from the GFF3 file of the TKS genome as a way to explore the replication types of its family members. The collinearity relationship between the five species, namely *T*. *kok-saghyz*, *A*. *thaliana*, *H*. *annuus*, *H*. *brasiliensis*, *L*. *sativa,* and *T*. *mongolicum*, was analyzed by combining their genome sequence information. Collinear relationships were obtained through MCScanX, and all collinear relationships were displayed through the Dual Systeny Plot function in the TBtools software (version v2.056) [68]. The Ka/Ks value is calculated using the Simple Ka/Ks Calculator function in the TBtools software (version v2.056). Genome downloads and identification results of related species are available in Appendix A. The PlantCARE database (http://bioinformatics.psb.ugent.be/webtools/plantcare/html/, accessed on 7 November 2023) was used to predict cis-acting elements of 2000 bp sequences upstream of each promoter of *TkHMGR* genes [69].

### 4.5. Plant Growth, Treatment, RNA Extraction, and qRT-PCR Analysis

The TKS seeds were obtained from Shihezi City, Xinjiang, China. When the seeds were six months old, the plants were selected and treated with hormones via Hoagland’s culture solution. The roots of TKS were irrigated with 100 μmol/L ETH and 1 mmol/ MeJA, and different tissue samples were collected at different periods of 0 h, 3 h, 6 h, 12 h, and 24 h, respectively. Three biological replicates were performed for each treatment, and the samples were subjected to liquid nitrogen snap-freezing or RNA extraction immediately after collection. The cDNA was obtained by using a difficult-to-extract total RNA kit (Megan, Guangzhou, China) and reverse transcription kit (Vazyme, Nanjing, China). The primer was designed with Primer Premier 5.0 software (https://premierbiosoft.com/, accessed on 20 November 2023) (Appendix A). Using *Tkβ-actin* as internal reference gene, the entire 20 µL qRT-PCR system consisted of 10 µL SYBR Green Master Mix (Vazyme, Nanjing, China), 1.3 µL diluted cDNA, 0.6 µL primers, and 7.5 µL ddH_2_O. The qRT-PCR reaction program was performed by LightCycler 480 II (Roche, Shanghai, China). The final results were processed via the 2^−ΔΔCt^ method, and the data were presented as mean ± standard deviation (SD), analyzed, and plotted based on a Student’s *t*-test (n = 3, * *p* < 0.05) using GraphPad Prism (version 9.5.1).

### 4.6. RNA-Seq Analysis and Determination of Plant Physiological Indicators

RNA-seq raw data of TKS roots at different developmental stages were obtained from NCBI public data (https://www.ncbi.nlm.nih.gov/sra/, accessed on 20 November 2023) (Accession: PRJNA539838). Then, the data were converted into fastq files with SRA-Toolkit v2.9 (NCBI, Bethesda, MD, USA). Raw reads were trimmed using Trimmomatic-0.39 [70]. The gene expression level was determined by mapping cleaned reads to the corresponding TKS reference genomes using the StringTie v2.1.3 package (GitHub, San Francisco, CA, USA) [71]. Gene expression levels were calculated according to the log2 (TPM + 1) values. Heat maps were plotted using TBtools software (version v2.056) with averaged TPM values from three replications.

The proline content, soluble sugar content, and chlorophyll content were measured using the Solarbio Physiological Indicator Assay Kit (Solarbio, Beijing, China), Proline (Pro) Content Assay Kit (Solarbio, Beijing, China), Plant Soluble Sugar Content Assay Kit (Solarbio, Beijing, China), and Chlorophyll Assay Kit (Solarbio, Beijing, China). The results were presented as mean ± standard deviation (SD), analyzed, and plotted based on Student’s *t*-test (n = 3, * *p* < 0.05) using GraphPad Prism (version 9.5.1).

### 4.7. Subcellular Localization of TkHMGRs

*TkHMGR* gene members’ full-length CDS removal stop codons (Appendix A) were subcloned into the Sma I/Xba I double enzyme digestion pCAMBIA1300-eGFP plasmid vector via homologous recombination and then transformed into *Agrobacterium tumefaciens* GV3101 via the freeze–thawing method. The endoplasmic reticulum localization plasmid marker pCAMBIA1300-35S-WAK2-mCherry-HDEL (Pyeast Biotechnology, Wuhan, China) was also transformed into *Agrobacterium tumefaciens* GV3101. *Agrobacterium tumefaciens* strains carrying recombinant plasmids were grown in liquid LB medium containing kanamycin (50 mg/L) and rifampicin (25 mg/L). The cells were collected and re-suspended in an infection solution (OD600 = 0.6) containing 10 mM MgCl_2_, 10 mM MES, and 100 µM acetosyringone. *Agrobacterium tumefaciens* containing a fusion expression vector and endoplasmic reticulum marker vector was mixed at a ratio of 1:1 and avoid light for 1 h, then injected into approximately 4-week-old tobacco leaves, and then a dark culture was performed for 72 h. Using Nikon AXR (confocal laser scanning microscope, CLSM) for final observation, the GFP excitation wavelength and emission wavelength were 488 nm and 500–550 nm, respectively, while the RFP excitation wavelength and emission wavelength were 561 nm and 570–620 nm, respectively. The fluorescence intensity of ROI was analyzed using ImageJ software (version 1.54g) and then plotted using GraphPad Prism (version 9.5.1).

## 5. Conclusions

In this study, a total of six members of the *HMGR* gene family were identified in *T. kok-saghyz,* and their basic physicochemical properties, gene structure, conserved domain, protein secondary structure, protein three-dimensional structural models, chromosome localization, collinearity, and phylogenetics were analyzed. TkHMGRs are highly conserved in the evolutionary process, and all six members retain the complete HMG-CoA reductases I conserved domain. We also explored the expression patterns of *TkHMGRs* under exogenous hormone ETH and MeJA treatments and the expression patterns of different stages of TKS root development. The subcellular localization results show that the six members are all located in the endoplasmic reticulum. Our findings will provide a theoretical basis for further understanding of the regulation of natural rubber biosynthesis, as well as theoretical support for subsequent functional verification and *T. kok-saghyz* breeding.

## Figures and Tables

**Figure 1 plants-13-02646-f001:**
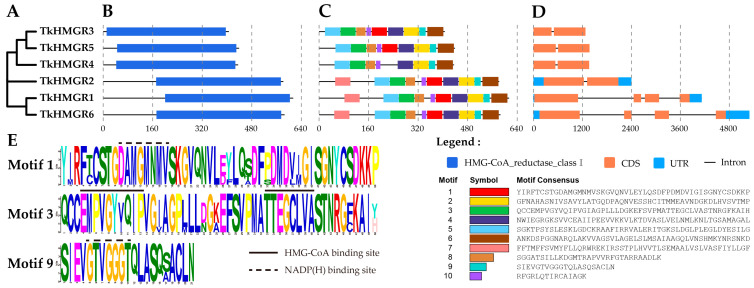
Phylogenetic relationship, conserved domain, motif analysis, and gene structure of the *TkHMGR* family members in *T. kok-saghyz*. (**A**) A phylogenetic tree based on the TkHMGR protein sequences was generated with MEGA 11.0 using the maximum likelihood (ML) method with 1000 bootstrap replications. (**B**) Predicted the distribution of conserved domains in TkHMGR proteins. (**C**) The top ten conserved motifs distribution of TkHMGR proteins; the color boxes represent different conserved motifs, as shown in the scheme on the lower right side of the figure. (**D**) The exon–intron structures of *TkHMGR* genes. (**E**) Three conserved motif logos including HMG-CoA binding sites and NADP(H) binding sites. Amino acids are represented by one-letter codes and presented in different colors.

**Figure 2 plants-13-02646-f002:**
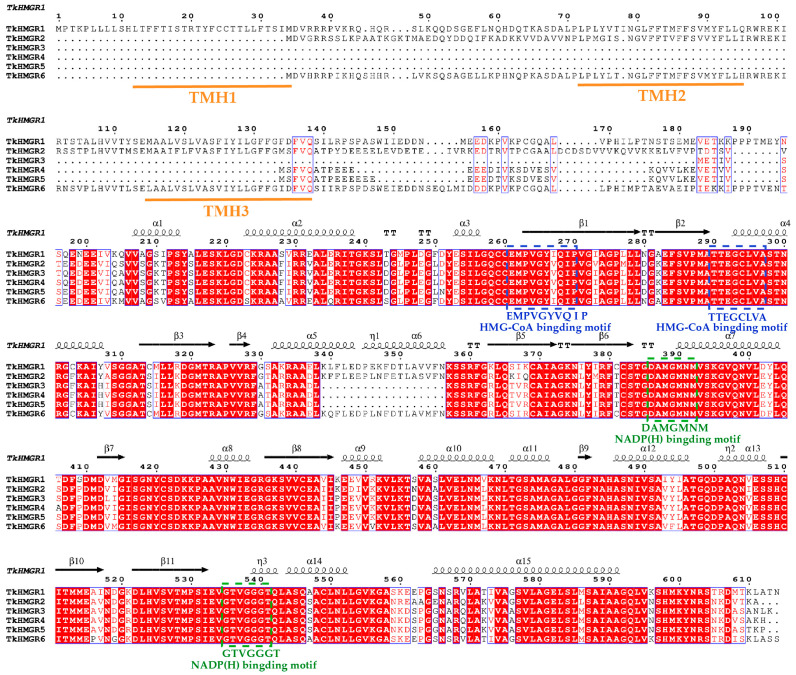
Multiple sequence alignment of the TkHMGR proteins. The secondary structural elements predicted with TkHMGR1 are shown above. Red box, white character represents strict identity; red character represents similarity in a group; blue frame represents similarity across groups. The η symbol refers to 3_10_-helixs; α-helixs and 3_10_-helixs are displayed as medium and small squiggles, respectively. β-strands are rendered as arrows; strict β-turns are denoted as TT letters. The orange line represents three different transmembrane helices. The blue and green boxes represent two HMG-CoA binding sites and two NADP(H) binding sites, respectively.

**Figure 3 plants-13-02646-f003:**
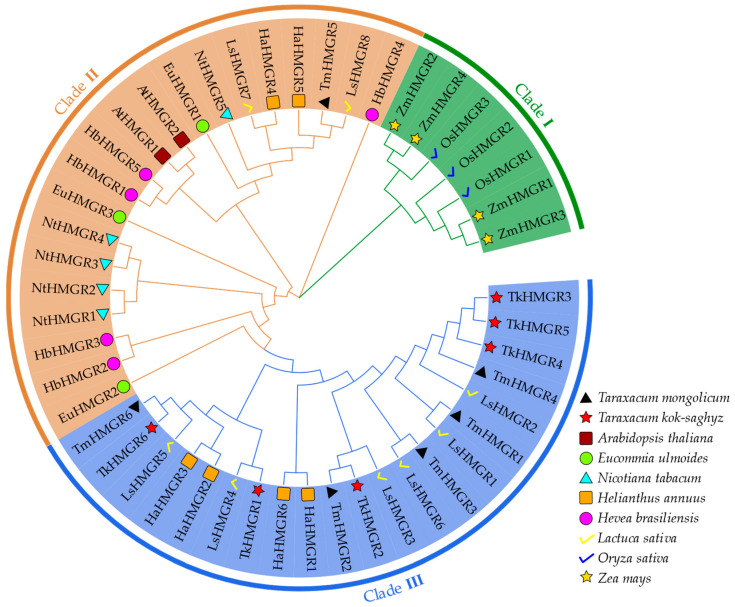
Phylogenetic analysis of HMGR proteins in *T. mongolicum*, *T. kok-saghyz*, *A. thaliana*, *E. ulmoides*, *N. tabacum*, *H. annuus*, *H. brasiliensis*, *L. sativa*, *O. sativa*, and *Z. mays*. The tree was constructed with MEGA 11.0 using the maximum likelihood (ML) method with 1000 bootstrap replications.

**Figure 4 plants-13-02646-f004:**
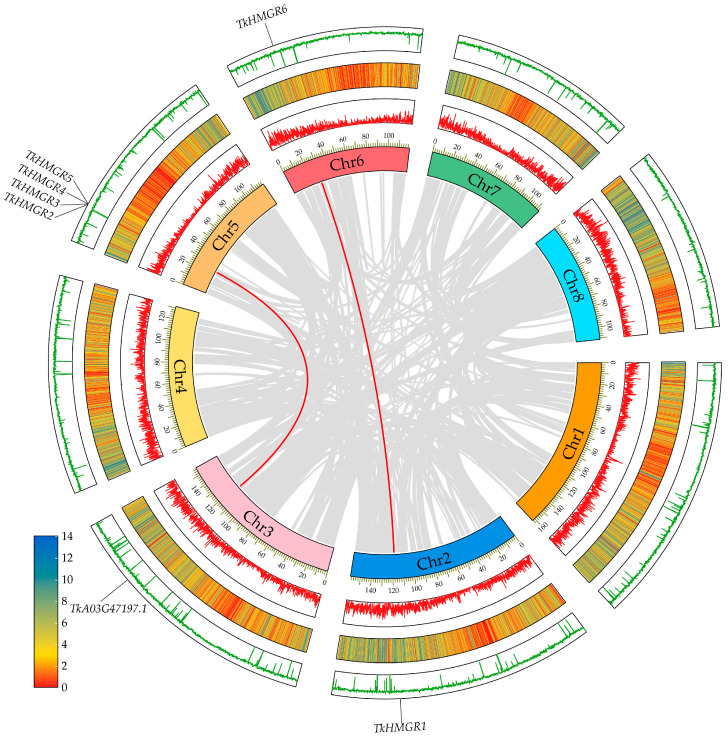
Collinearity analysis of *TkHMGR* genes. The red lines show the *TkHMGR* gene pairs replicated; the grey lines indicate collinear blocks across the whole genome. The innermost track of the Circos plot indicates the chromosome length and number. The second and third tracks indicate the density of genes on the corresponding chromosome; the gene density distribution of the TKS genome varies from 0 to 14, with an average of 4 genes per 100 kb. The outermost track represents the corresponding GC content of the whole genome.

**Figure 5 plants-13-02646-f005:**
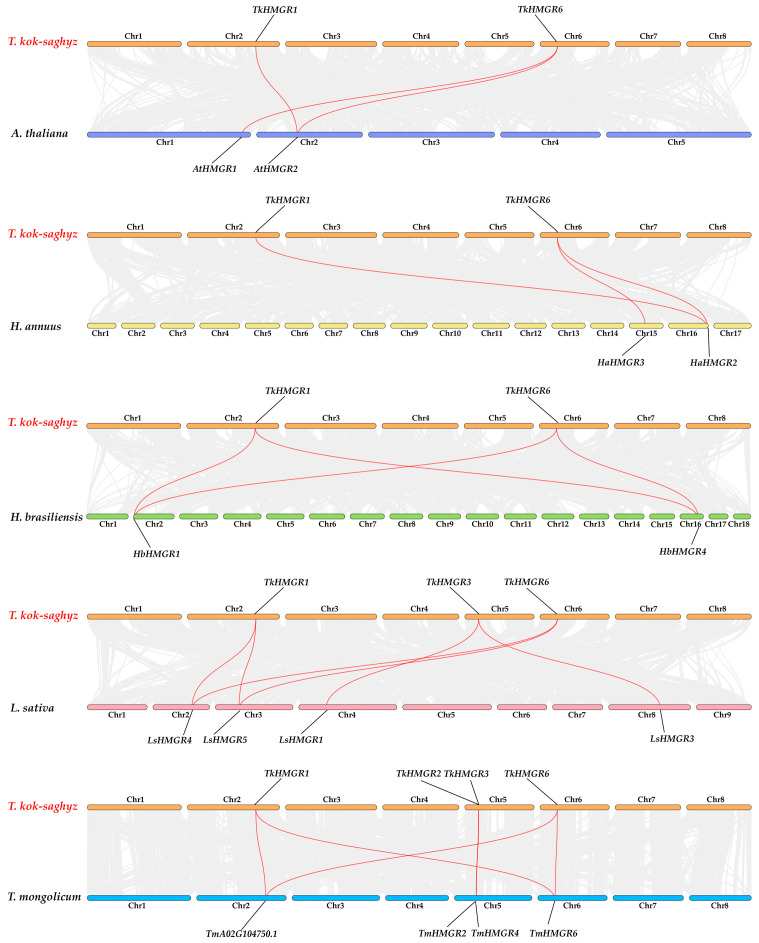
Collinearity analysis of *HMGR* genes between *T. kok-saghyz*, *A. thaliana*, *H. annuus*, *H. brasiliensis*, *L. sativa*, and *T. mongolicum*. The gray lines represent homologous gene pairs between TKS and other species. The highlighted red lines represent the collinear relationship of *HMGR* gene pairs.

**Figure 6 plants-13-02646-f006:**
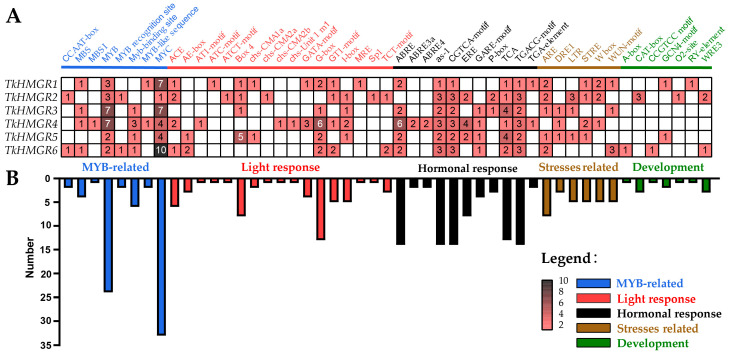
Predicted the cis-acting elements of the promoters 2000 bps upstream of *TkHMGR* genes. (**A**) The number of cis-acting elements in different classifications of each gene. (**B**) The total number of different cis-acting elements.

**Figure 7 plants-13-02646-f007:**
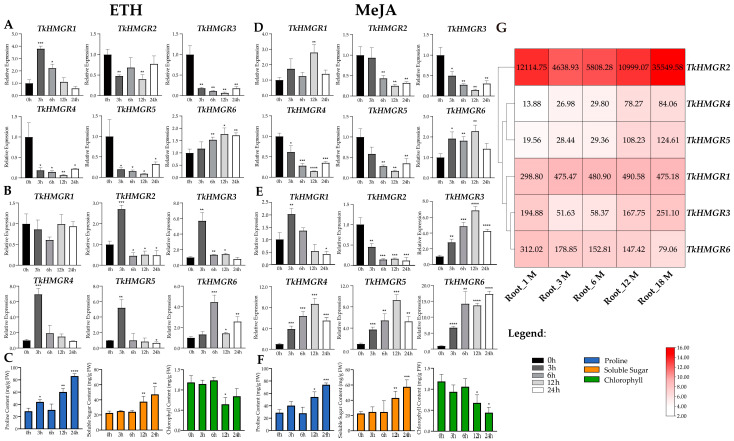
Expression profiles of *TkHMGR* genes and the changes in proline, soluble sugar, and chlorophyll content under exogenous hormone treatment. (**A**) Expression levels in roots of six-month-old TKS treated with 100 μmol/L ETH. (**B**) Expression levels in leaves of six-month-old TKS treated with 100 μmol/L ETH. (**C**) Proline and soluble sugar content in 6-month-old TKS roots treated with ETH; chlorophyll content in 6-month-old TKS leaves treated with ETH. (**D**) Expression levels in roots of six-month-old TKS treated with 1 mmol/L MeJA. (**E**) Expression levels in leaves of six-month-old TKS treated with 1 mmol/L MeJA. (**F**) Proline and soluble sugar content in 6-month-old TKS roots treated with MeJA; chlorophyll content in 6-month-old TKS leaves treated with MeJA. Data are shown as the means ± SD of three independent biological repetitions using the 2^−ΔΔCt^ method with *Tkβ-actin* as the internal standard for normalization. Asterisks represent significant differences via the Student’s *t*-test analysis compared with the control (* *p* < 0.05, ** *p* < 0.01, *** *p* < 0.001, and **** *p* < 0.0001). (**G**) The heatmap display of TKS roots from different developmental stages: 1 month old, 3 months old, 6 months old, 12 months old, and 18 months old based on transcript per million (TPM) values. Red indicates high expression levels, and white indicates low expression levels. The numbers on the box represent the TPM value.

**Figure 8 plants-13-02646-f008:**
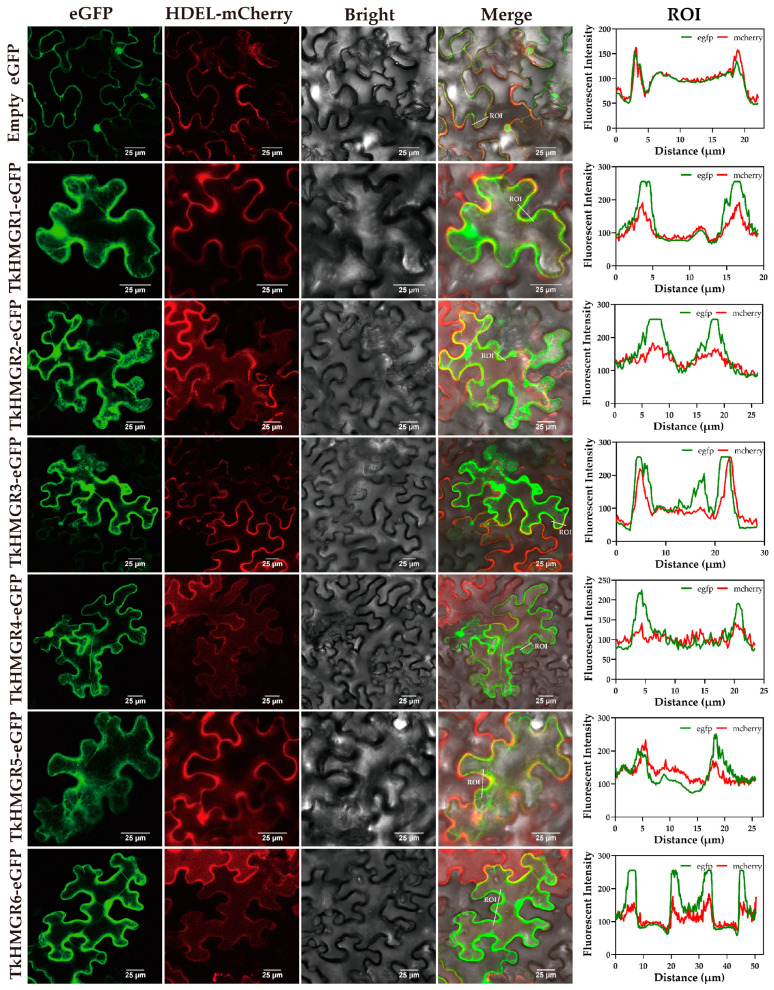
The subcellular localization of six TkHMGR proteins. The empty eGFP vector was used as the control. Endoplasmic reticulum (ER) marker was labeled with mCherry. The white line represents the region of interest in the merge field. Bar = 25 μm.

**Table 1 plants-13-02646-t001:** The detail information of *HMGR* gene family members in *T. kok-saghyz*.

Gene Name	Gene ID ^1^	CDS Length (bp)	Protein Length (aa)	MW (kDa) ^2^	pI ^3^	Predicted Location(s)	Number of Predicted TMHs ^4^
*TkHMGR1*	*TkA02G464040*	1845	614	66.688	8.05	Endoplasmic reticulum	3
*TkHMGR2*	*TkA05G087240*	1749	582	62.642	5.58	Endoplasmic reticulum	2
*TkHMGR3*	*TkA05G087830*	1221	406	42.694	7.50	Endoplasmic reticulum	0
*TkHMGR4*	*TkA05G087870*	1311	436	46.061	5.68	Endoplasmic reticulum	0
*TkHMGR5*	*TkA05G087940*	1320	439	46.443	5.32	Endoplasmic reticulum	0
*TkHMGR6*	*TkA06G113340*	1761	586	62.902	6.65	Endoplasmic reticulum	2

^1^ Gene ID is based on the latest released version of TKS genome [37]. ^2^ Molecular weight. ^3^ Isoelectric point. ^4^ Transmembrane helices.

**Table 2 plants-13-02646-t002:** The secondary structure of HMGR proteins in *T. kok-saghyz*.

Protein Name	α-Helix	β-Turn	Extended Strand	Random Coil
TkHMGR1	243 (39.58%)	36 (5.86%)	96 (15.64%)	239 (38.93%)
TkHMGR2	257 (44.16%)	38 (6.53%)	94 (16.15%)	193 (33.16%)
TkHMGR3	197 (48.52%)	27 (6.65%)	70 (17.24%)	112 (27.59%)
TkHMGR4	213 (48.85%)	23 (5.28%)	69 (15.83%)	131 (30.05%)
TkHMGR5	215 (48.97%)	25 (5.69%)	69 (15.72%)	130 (29.61%)
TkHMGR6	252 (43%)	30 (5.12%)	81 (13.82%)	223 (38.05%)

**Table 3 plants-13-02646-t003:** The gene duplication type analysis of *HMGR* gene family members in *T. kok-saghyz*.

Gene ID	Duplication Type(s)
*TkHMGR1*	WGD or Segmental
*TkHMGR2*	WGD or Segmental
*TkHMGR3*	Tandem
*TkHMGR4*	Tandem
*TkHMGR5*	Tandem
*TkHMGR6*	WGD or Segmental

## Data Availability

All data generated or analyzed during this study are included in this article.

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
