# Peer review of "Genome-Wide Identification and Characterization of the HMGR Gene Family in Taraxacum kok-saghyz Provide Insights into Its Regulation in Response to Ethylene and Methyl Jsamonate Treatments"

_plants, 2024, doi:10.3390/plants13182646_

Round 1

Reviewer 1 Report

Comments and Suggestions for Authors

The manuscript presents a comprehensive study on the TkHMGR gene family in Taraxacum, with a focus on their role in natural rubber biosynthesis. The paper is well-organized and provides substantial data on gene identification, phylogenetic relationships, conserved structures, and expression patterns. The research addresses a significant problem, the global shortage of natural rubber, and offers valuable insights into potential alternative sources and genetic strategies to enhance rubber production.

1.        Gene Structure and Expression Analysis: The approach to analyzing gene structure, protein motifs, and expression patterns is robust. More details on the software parameters used would enhance reproducibility.

2.        The potential applications in genetic engineering and molecular breeding are well-articulated. Suggestions for future research could be expanded, especially regarding functional validation of candidate genes.

Author Response

Response to Reviewer 1:

Comments 1:

  1.        Gene Structure and Expression Analysis: The approach to analyzing gene structure, protein motifs, and expression patterns is robust. More details on the software parameters used would enhance reproducibility.

  1.        The potential applications in genetic engineering and molecular breeding are well-articulated. Suggestions for future research could be expanded, especially regarding functional validation of candidate genes.

Response 1: Thank you for your valuable feedback and affirmation of the manuscript, which has improved the readability and scientificity of the revised manuscript.

Reviewer 2 Report

Comments and Suggestions for Authors

The manuscript entitled “Genome-Wide Identification of the HMGR Gene Family in Taraxacum kok-saghyz Provide Insights into Its Expression Patterns in Response to Ethylene and Methyl Jsamonate Treatments” presents a detailed study of the TkHMGR gene family on genomic and transcriptional level, but the authors completely neglect that HMGR gene identification and expression data in T. koksaghyz have been already published. Furthermore, two major experiments presented in the second part are not reliable in my opinion (see major remarks below) so that the authors have to repeat or extend their experimental approaches before I would consider the manuscript for publication.

Major remarks: 

- The authors have to clarify that their study is not the first one regarding the identification of HMGRs from T. koksaghyz. The identification of 8 HMGR genes and corresponding expression data was already published by Lin et al (2022). The authors should refer/comment on this publication and the presented data and they should use or adjust their nomenclature and numbering of genes. Additionally, the already published expression data presented by Lin et al. should be included in their gene classification and characterization.

- Hormone treatment experiment: Pleases specify the age of the plant during the hormone treatment/ at harvesting time after treatment. Six-months old plants (as mentioned in Figure 7) are fully developed plants with fully developed roots. How did the authors treat the roots comparably? A hormone treatment in seedling stage or with younger plants (up to 8-week old plants) would be more reliable. Furthermore, I do not believe that 6-month-old plants accumulate after 3h or 24h after treatment up to 10% NR compared to the control. This cannot be an effect of metabolic activity during a time-period of 3 or 24 hours in fully developed plants. Additionally, the Alkali Boiling Method used is not a state-of-the-art method for NR quantification as described also in their cited paper [66] “Rough estimate method with high variance”. The authors have to present comprehensive data using reliable methods.

- Subcellular Localization of TkHMGR, TkHMGR2, TkHMGR6: Line 500-501: Please specify the ER plasmid marker; Line 509-510: please mention the emission wavelengths (in addition to excitation); In Line 361 (discussion part) the authors stated that they found “that all members of TkHMGRs gene family in T. kok-saghyz were located on the endoplasmic reticulum”. This is only the result of the predicted location based on sequence analysis. If the authors want to make statements about the subcellular localization of the different HMGRs, all HMGRs should be included in the infiltration experiments shown in Fig. 8. Furthermore, the localization of HMGR3, 4 and 5 should be presented as control in Fig. 8. to be sure that the presented co-localisation for the other three HMGRs is reliable. In case of e.g. TkHMGR6 the mCherry signal is too weak compared to the neighboring cell, so that no co-localisation of the two fluorophores is visible. To proof co-localisation quantitative analysis of selected regions of interest (ROIs) has to be presented.

Minor remarks:

- I would recommend to choose another title for the manuscript, because there is no direct connection between “identification of…” and “provide insights into its expression pattern…”. My title recommendation: “Genome-Wide Identification and Characterisation of the HMGR Gene Family in Taraxacum kok-saghyz Provide Insights into Its Regulation in Response to Ethylene and Methyl Jsamonate Treatments

- line 25: change “are” to “is” the most prominent feature

- line 26: explain the abbreviation WGD in the manuscript

- line 55: please check your reference. In which of these paper a range of 2.89% to 27.89% is shown for the NR content?

- line 97-99: please read the literature you cited more carefully and cite correctly. In the mentioned paper [32] the authors overexpress an HMGR from Arabidopsis in dandelion to increase the triterpene content. The cis-isoprene content (not transcription level) was upregulated by co-expression of two additional enzymes.

- Figure 8: change “mCheery” to “mCherry”

- Line 259: please provide sequence data for TmA02G104750.1

Author Response

Response to Reviewer 2:

Thank you for pointing out these problems and the highly valuable comment.

Comments 1: 

-The authors have to clarify that their study is not the first one regarding the identification of HMGRs from T. koksaghyz. The identification of 8 HMGR genes and corresponding expression data was already published by Lin et al (2022). The authors should refer/comment on this publication and the presented data and they should use or adjust their nomenclature and numbering of genes. Additionally, the already published expression data presented by Lin et al. should be included in their gene classification and characterization.

Response 1: The authors are very sorry for our careless mistakes. Thank you for pointing this out, we agree with this comment. As suggested by the reviewer, we must clarify we are not the first to identification of HMGRs in TKS, this mistake has been corrected in the revised manuscript, as seen in page 4, line 150, Table 1; page 14, line 376-384; page 15, line 442-443; Table S7 and Table S8. The article published by Lin et al (2022) has preliminarily identified the genes related to the natural rubber synthesis pathway and explored their expression levels in different tissues of TKS. And our article is a more comprehensive and systematic analysis of TkHMGRs gene family based on their foundation. Our study identified six family members using blastp and conserved domain identification methods in the TKS genome, which is different from what Lin identified. The determination of gene family members is the most important thing in family analysis, by comparing the identified members, we noticed that the two unidentified members were annotated as pseudogenes in the genome annotation GFF3 file and did not have corresponding protein sequences in the proteome, we have added these findings in the discussion section (page 14, line 376-384) of the manuscript, and provided supporting materials Table S7 and S8 (can see below). Except for the two unidentified we renamed the genes is also based on Lin's published. We also discussed the expression patterns of different developmental stages analyzed with Lin's publication (page 15, line 442-443). Thanks reviewer again to point out these details, we had tried our best to made changes and corrected in the manuscript.

Comments 2: 

-Hormone treatment experiment: Pleases specify the age of the plant during the hormone treatment/ at harvesting time after treatment. Six-months old plants (as mentioned in Figure 7) are fully developed plants with fully developed roots. How did the authors treat the roots comparably? A hormone treatment in seedling stage or with younger plants (up to 8-week old plants) would be more reliable. Furthermore, I do not believe that 6-month-old plants accumulate after 3h or 24h after treatment up to 10% NR compared to the control. This cannot be an effect of metabolic activity during a time-period of 3 or 24 hours in fully developed plants. Additionally, the Alkali Boiling Method used is not a state-of-the-art method for NR quantification as described also in their cited paper [66] “Rough estimate method with high variance”. The authors have to present comprehensive data using reliable methods.

Response 2: Thank you for your insightful suggestion, we have taken all these comments and suggestions into account and have made major corrections in the revised manuscript.  We have corrected the shortcomings and clearly stated the plant age and harvest time on page 10, line 301-303.

-As the reviewer pointed out, the six-month-old plants with fully developed roots, but during this period, the TKS root system developed relatively uniformly (doi: 10.1016/j.indcrop.2022.114776, Figure 1A), the main roots and nature rubber content are the focus of our attention, in future research, we will also focus on 6-month-old plants so we chose the 6-month-old plants to do hormone treatment experiment.

-The method (Alkali Boiling Method) to determination nature rubber content is indeed very rough and unreliable, and we highly agree with reviewer. Therefore, we have modified the method to use infrared spectroscopy and conducted additional experiments (determination of changes in chlorophyll, root proline content, and root soluble sugar content of TKS after hormone treatment), and added the expression profile of TkHMGRs in TKS leaves and their expression at different root development stages to make the experiment more reliable (page 10, results 2.6, Figure 7, line 298-351 and Figure S3) .

Comments 3: - Subcellular Localization of TkHMGR, TkHMGR2, TkHMGR6: Line 500-501: Please specify the ER plasmid marker; Line 509-510: please mention the emission wavelengths (in addition to excitation); In Line 361 (discussion part) the authors stated that they found “that all members of TkHMGRs gene family in T. kok-saghyz were located on the endoplasmic reticulum”. This is only the result of the predicted location based on sequence analysis. If the authors want to make statements about the subcellular localization of the different HMGRs, all HMGRs should be included in the infiltration experiments shown in Fig. 8. Furthermore, the localization of HMGR3, 4 and 5 should be presented as control in Fig. 8. to be sure that the presented co-localisation for the other three HMGRs is reliable. In case of e.g. TkHMGR6 the mCherry signal is too weak compared to the neighboring cell, so that no co-localisation of the two fluorophores is visible. To proof co-localisation quantitative analysis of selected regions of interest (ROIs) has to be presented.

Response 3: Thank you for your comments and suggestions concerning our manuscript, and we have also taken your suggestion. The ER marker plasmid and the excitation /emission wavelength has been specifically explained (The endoplasmic reticulum marker created by combining the signal peptide of At-WAK2 at N-terminus of the mcherry and the HDEL (his-asp-glu-leu) endoplasmic re-ticulum retention signal at C-terminus) (The final observation in Nikon AXR (confocal laser scanning microscope, CLSM), the excitation and emission wavelength of GFP or RFP is 488 nm and 561 nm or 500-550nm and 570-620 nm, respectively.) (page 12, linr354-356; page 17, line 546-548; line 555-557). We also added experiments on subcellular localization of TkHMGR3, TkHMGR4, and TkHMGR5 (page 13, Figure 8).

Comments 4: - I would recommend to choose another title for the manuscript, because there is no direct connection between “identification of…” and “provide insights into its expression pattern…”. My title recommendation: “Genome-Wide Identification and Characterisation of the HMGR Gene Family in Taraxacum kok-saghyz Provide Insights into Its Regulation in Response to Ethylene and Methyl Jsamonate Treatments”

Response 4: Thank you for this valuable suggestion. We have taken your suggestion and revised the title of the manuscript.

Comments 5: - line 25: change “are” to “is” the most prominent feature

- line 26: explain the abbreviation WGD in the manuscript

Response 5: Thanks to the reviewer for this careful comment. The author has made revisions on line 25 of the manuscript and explained the WGD full name (line 26).

Comments 6: - line 55: please check your reference. In which of these paper a range of 2.89% to 27.89% is shown for the NR content?

- line 97-99: please read the literature you cited more carefully and cite correctly. In the mentioned paper [32] the authors overexpress an HMGR from Arabidopsis in dandelion to increase the triterpene content. The cis-isoprene content (not transcription level) was upregulated by co-expression of two additional enzymes.

Response 6: We would like to thank the reviewer for pointing out this issue. The authors have checked the reference and correctly cited it (page 1, line 55-56; page 2, line 98-100).

Comments 7: - Figure 8: change “mCheery” to “mCherry”

- Line 259: please provide sequence data for TmA02G104750.1

Response 7: Thank you for kindly reminding us. We are sorry for our careless mistakes. The spelling error has been corrected (page 13, Figure8, below). And we provided sequence data for TmA02G104750.1 in Table S3 of Supporting Materials.

Reviewer 3 Report

Comments and Suggestions for Authors

In this study, Du and colleagues identified and analyzed six members of the TkHMGRs gene family in Taraxacum kok-saghyz. Various bioinformatics and in vivo approaches were employed, including gene and protein structure analysis, subcellular localization, chromosome localization, collinearity, and phylogenetic relationships. The expression patterns of the six TkHMGRs under exogenous hormone ETH and MeJA treatments were also examined in relation to natural rubber biosynthesis regulation.

The bioinformatics analyses are extensive and comprehensively conducted. However, the work's second part, concerning hormonal regulation and its effect on NR production, which is more interesting from an application perspective, remains preliminary and is not clearly discussed. Here are my main concerns and suggestions:

  • The phylogenetic analysis of HMGR proteins in several plant species should be the first analysis shown and described, providing a more general overview before moving to the detailed analysis of the six members identified in Taraxacum kok-saghyz. The authors should explain why TkHMGR3 clusters with TkHMGR5 in the phylogenetic tree in Figure 3, while in the phylogenetic analysis shown in Figure 1A, TkHMGR5 is closer to TkHMGR4. Perhaps adding an outgroup could improve the phylogenetic analysis in Figure 1A.
  • The legend of Figure 1C lacks a brief description of the different motifs.
  • In the analysis of predicted cis-acting elements, showing the sum of the elements identified in the six different promoter regions does not seem logical or informative. It might be better to discuss the peculiarities of each promoter. For instance, TkHMGR6 has a high frequency of MYC elements. It seems that the ETH treatment was chosen based not on cis-element prediction but on existing literature indicating that ETH enhances plant growth and NR production.
  • In the expression analysis with ETH and MeJA treatments, TkHMGR1 and TkHMGR6 are upregulated while all others are downregulated. The authors suggest that TkHMGR1 and TkHMGR6 might play a major role. However, an important piece of missing information is the comparison of the expression levels of the six TkHMGRs in roots under normal conditions. Which TkHMGRs are normally more expressed in roots? Are TkHMGR1 and TkHMGR6 the most expressed in root tissues? Are transcriptomic data available to verify this? Alternatively, could the authors perform quantitative gene expression analysis such as droplet digital PCR or tissue-specific expression pattern analysis like in-situ hybridization? 
  • Which ER marker was used? the HDEL-mCherry?
Comments on the Quality of English Language

 Minor editing of English language required

Author Response

Response to Reviewer 3:

The authors would like to thank the reviewer for their constructive comments which have enabled us to produce a significantly improve manuscript.

Comments 1: The phylogenetic analysis of HMGR proteins in several plant species should be the first analysis shown and described, providing a more general overview before moving to the detailed analysis of the six members identified in Taraxacum kok-saghyz. The authors should explain why TkHMGR3 clusters with TkHMGR5 in the phylogenetic tree in Figure 3, while in the phylogenetic analysis shown in Figure 1A, TkHMGR5 is closer to TkHMGR4. Perhaps adding an outgroup could improve the phylogenetic analysis in Figure 1A.

Response 1: Thank you for your valuable suggestions. The suggestions for placing the phylogenetic analysis as the first result, this completely feasible. But the authors would like to stay as is, because in the Results 2.4 part we carried out collinearity analysis, the species which chose to Figure 5 is according to the phylogenetic of HMGR proteins in several plants species. Results 2.3 multi species phylogenetic analysis can be combined with the collinearity of Results 2.4, that seem making the manuscript more logical. And we very sincerely appreciate the valuable comments about ‘’why TkHMGR3 clusters with TkHMGR5 in the phylogenetic tree in Figure 3, while in the phylogenetic analysis shown in Figure 1A, TkHMGR5 is closer to TkHMGR4’’, it was your professional review work on our manuscript that made us realize we had a big mistake. The reason why TkHMGR5 not clusters with TkHMGR3 is our due to our negligence in placing the wrong phylogenetic relationship (constructed by NJ method), while our manuscript wants to display is constructed by ML method. At first, we tried these two methods to analyze evolutionary relationships, and the author has corrected the mistake, put the correct Figure (page 4, Figure 1). In the revised figure, TkHMGR3~TkHMGR5 are clustered together, thank you again for the careful reminder from the reviewer.

Comments 2: The legend of Figure 1C lacks a brief description of the different motifs.

Response 2: Thank you for your helpful suggestion. The authors have added the brief description (page 4, line 176-178)

Comments 3: In the analysis of predicted cis-acting elements, showing the sum of the elements identified in the six different promoter regions does not seem logical or informative. It might be better to discuss the peculiarities of each promoter. For instance, TkHMGR6 has a high frequency of MYC elements. It seems that the ETH treatment was chosen based not on cis-element prediction but on existing literature indicating that ETH enhances plant growth and NR production.

Response 3: Thank you for your comments that help improve our manuscript. We did indeed overlook other information; the authors have made revised based on your feedback and added it to abstract (page 1, line 29; page10 290-295). (On the other hand, the white values inside the small box significantly indicate that the gene has a higher number of this element (Figure 6A). We found that TkHMGR1 had 7 MYC elements; TkHMGR3 had 7 MYB elements and 7 MYC elements; TkHMGR4 had 7 MYB elements, 6 G-Box elements and 6 ABRE elements; TkHMGR5 had 5 Box4 ele-ments; TkHMGR6 had the most MYC-element, with 10. These results reveal that the drought response of TkHMGRs may be mainly regulated by MYB and MYC transcrip-tion factors.)

Comments 4: In the expression analysis with ETH and MeJA treatments, TkHMGR1 and TkHMGR6 are upregulated while all others are downregulated. The authors suggest that TkHMGR1 and TkHMGR6 might play a major role. However, an important piece of missing information is the comparison of the expression levels of the six TkHMGRs in roots under normal conditions. Which TkHMGRs are normally more expressed in roots? Are TkHMGR1 and TkHMGR6 the most expressed in root tissues? Are transcriptomic data available to verify this? Alternatively, could the authors perform quantitative gene expression analysis such as droplet digital PCR or tissue-specific expression pattern analysis like in-situ hybridization? 

 Response 4: Thank you honorable reviewer for your comments. We carried out RNA-seq analysis, and in the revised manuscript, the expression levels of six genes under normal conditions have been added and discussed (page 11, Figure 7). we also added the leaves expression levels (page 11, Figure 7; Figure S3). The authors agree with the reviewer that more experimental results would be useful to understand the expression levels of genes. However, limited by the laboratory conditions, it is impractical to implement the related experiments. In other side, we also added physiological indexes experimental and other gene’s subcellular localization in our revised manuscript (page 10, results 2.6; page 12, results 2.7). In the future, the authors would pay more attention to related studies for deeply and thoroughly understand this problem. Thank you very much for this creative idea.

Comments 5: Which ER marker was used? the HDEL-mCherry?

Response 5: We would like to thank the reviewer for pointing out this issue. The ER marker plasmid has been specifically explained (page 12, linr356-358; page 17, line 542-543; line 550-553).

Round 2

Reviewer 2 Report

Comments and Suggestions for Authors

Thanks for providing a revised version of the manuscript. However, even though the authors included additional data the two major experiments presented in the second part are still not reliable in my opinion (see response 2 and 3 below). Therefore, I recommend to reject the paper.

Reviewer response 2:

Thanks for providing additional information and data for the hormone treatment experiment, but I have still questions and doubts regarding the methodology: Did the authors harvest the whole root after the hormone treatment at each time-point and for three biological replicates? How did the authors exclude a biological variance in the material? As shown in Xie et al. (2022) doi:10.1016/j.indcrop.2022.114776, the NR variance in 6-month-old roots is between 5.0 and around 6.2% and the data shown by the authors did not exceed this natural variance occurring in heterogenous Tk-plants. Furthermore, in case of ETH-treatment the NR decreases after 12 hours. How do the authors explain this effect, if they stated that treatment causes NR production (NR will not be naturally degraded in the material). In my opinion, the presented data do not provide enough evidence for NR- upregulation as a result of hormone treatment and/or an influence of HMGR expression.

Additionally, why do the authors present proline, sugar and chlorophyll data in the revised version? What is the relation between HMGR and those metabolites? The authors have to introduce, explain and discuss the relation throughout the manuscript and should explain the relevance of the metabolite data for their HMGR-study. Furthermore, why do the authors expect and measure chlorophyll in a non-photosynthetic tissue like the roots?

Reviewer response 3:

Thanks for providing additional confocal pictures for TkHMGR3, TkHMGR4 and TkHMGR5, but comparable to TkHMGR1 and TkHMGR6 those picture do not provide any evidence for ER-localisation of the TkHMGRs. As I mentioned in the first review in case of e.g. TkHMGR6 the mCherry signal is too weak compared to the neighboring cell, so that no co-localisation of the two fluorophores is visible. This is also the case for TkHMGR3 and 4. To proof co-localisation convincing and high resolution confocal pictures should be presented or quantitative analysis of selected regions of interest (ROIs) has to be presented as shown e.g for the ER-localisation of a rubber transferase activator in Taraxacum DOI: (10.1038/NPLANTS.2015.48)

Author Response

Response to Reviewer 2:

We would like to thank the reviewer for your professional review work, constructive comments, and valuable suggestions on our manuscript. As you are concerned, several issues need to be addressed, which are replied to in detail as below. And we have carefully revised the manuscript, we hope these changes will strengthen our manuscript.

Comments 1:

Thanks for providing additional information and data for the hormone treatment experiment, but I have still questions and doubts regarding the methodology: Did the authors harvest the whole root after the hormone treatment at each time-point and for three biological replicates? How did the authors exclude a biological variance in the material? As shown in Xie et al. (2022) doi:10.1016/j.indcrop.2022.114776, the NR variance in 6-month-old roots is between 5.0 and around 6.2% and the data shown by the authors did not exceed this natural variance occurring in heterogenous Tk-plants. Furthermore, in case of ETH-treatment the NR decreases after 12 hours. How do the authors explain this effect, if they stated that treatment causes NR production (NR will not be naturally degraded in the material). In my opinion, the presented data do not provide enough evidence for NR- upregulation as a result of hormone treatment and/or an influence of HMGR expression.

Response 1:

Thank you for the comments and suggestions on the manuscript. Your constructive comments have made us aware of the shortcomings of our approach to the effects of hormone treatment experiments on the yield of natural rubber. The reliability of our experiment was not sufficient to explain that hormone treatments cause changes in NR content, we apologize for this, and we decided to remove this part. Accumulation of natural rubber is a complex process, the effect of applying exogenous hormones on the rubber content is another story, and the content of our manuscript is not enough to explain this effect, we apologize again, and after that we will improve our methods to obtain more reliable experimental data. In addition, we provided expression profiles of the leaves and analysed the expression patterns of TkHMGRs. The aim is to provide more insights into the expression of the TkHMGRs gene family under ETH and MeJA treatments. (See Page 13, Figure7; Page 11, line326-328, line 334-339)

Comments 2:

Additionally, why do the authors present proline, sugar and chlorophyll data in the revised version? What is the relation between HMGR and those metabolites? The authors have to introduce, explain and discuss the relation throughout the manuscript and should explain the relevance of the metabolite data for their HMGR-study. Furthermore, why do the authors expect and measure chlorophyll in a non-photosynthetic tissue like the roots?

Response 2:

Thank the reviewer for raising these important points. The authors measure the proline, soluble sugars and leaf chlorophyll content to explore changes in plant physiological conditions under the treatment of these two exogenous hormones. Changes in these three indicators reflect to some extent the plant's stress tolerance. And based on your suggestion, we have also explained this part in the discussion section. (See Page 17, line 477-491)

Comments 3:

Thanks for providing additional confocal pictures for TkHMGR3, TkHMGR4 and TkHMGR5, but comparable to TkHMGR1 and TkHMGR6 those picture do not provide any evidence for ER-localisation of the TkHMGRs. As I mentioned in the first review in case of e.g. TkHMGR6 the mCherry signal is too weak compared to the neighboring cell, so that no co-localisation of the two fluorophores is visible. This is also the case for TkHMGR3 and 4. To proof co-localisation convincing and high resolution confocal pictures should be presented or quantitative analysis of selected regions of interest (ROIs) has to be presented as shown e.g for the ER-localisation of a rubber transferase activator in Taraxacum DOI: (10.1038/NPLANTS.2015.48)

Response 3:

Thank you for the valuable comments and provides relevant paper, we think this is an excellent suggestion. As suggested by the reviewer, we have revised the subcellular localisation figure, regions of interest were selected at each merge field for quantitative analysis to demonstrate co-localisation. We hope that our modifications will make the experimental results more convincing. (See Page 15, Figure 8; Page 19, line 612-614)

Reviewer 3 Report

Comments and Suggestions for Authors

I thank the authors for carefully considering all of my comments. I am pleased that my concern about the discrepancy between the two phylogenetic analyses helped to correct an error. Including RNA-seq data strengthens the identification of the roles of TkHMGR1,2 and 6. Overall, all the changes to the text clarify and assist the reader in understanding. I believe the manuscript has significantly improved after the revisions.

Author Response

We would like to thank you for your professional review work, constructive comments, and valuable suggestions on our manuscript. Your time and efforts are greatly appreciated.

Round 3

Reviewer 2 Report

Comments and Suggestions for Authors

Thanks for providing a revised version of the manuscript. Since the authors considered all recommendations and issues and changed the manuscript accordingly I recommend the manuscript for publication in the present form.